# Efficient fibre-pigtailed source of indistinguishable single photons

Nico Margaria [1], Florian Pastier[1], Thinhinane Bennour[1], Marie Billard[1], Edouard Ivanov[1], William Hease[1], Petr Stepanov[1], Albert F. Adiyatullin[1], Raksha Singla[1], Mathias Pont[1], Maxime Descampeaux [1], Alice Bernard[1], Anton Pishchagin[1], Martina Morassi[2], Aristide Lemaître [2], Thomas Volz[3], Valérian Giesz[1], Niccolo Somaschi [1] ✉, Nicolas Maring [1], Sébastien Boissier [1], Thi Huong Au[1] & Pascale Senellart[2]

Semiconductor quantum dots in microcavities are an excellent platform for the efficient generation of indistinguishable single photons. However, their use in a wide range of quantum technologies requires their controlled fabrication and integration in compact closed-cycle cryocoolers, with a key challenge being the efficient and stable extraction of the single photons into a single-mode fibre. Here we report on a method for the fibre-pigtailing of deterministically fabricated single-photon sources. Our technique allows for nanometre-scale alignment accuracy between the source and a fibre, alignment that persists all the way from room temperature to 2.4 K. We demonstrate high performance of the device under near-resonant optical excitation with a photon indistinguishability of 97.5 % and a brightness at the output fibre of the system of 20.8 %. We show that the indistinguishability and single-photon rate are stable for over ten hours of continuous operation in a single cooldown. We further confirm that the device performance is not degraded by nine successive cooldown-warmup cycles.

Recent advances in quantum photonics have led to the generation, processing, and detection of quantum states of light with increasing size and complexity[1,2]. The scalability of these protocols is limited by the challenge of generating identical single-photons with high efficiency. Single-photon sources based on semiconductor quantum dots (QDs) coupled to microcavities have demonstrated unparalleled efficiencies and single-photon purities[3–7], with no fundamental compromise to optimise both properties simultaneously. These devices have enabled fast progress in the field of photonic quantum computing[8–12] and photonic quantum communication[13–21]. Underpinning these recent successes is the ability to precisely control the coupling of single QDs to microcavities as a way to efficiently funnel the emitted photons into single-mode fibres[22].

A remaining obstacle to the widespread adoption of QD-based single-photon sources is the need to operate them at cryogenic temperatures around 5 K. Particularly, the requirements of optical excitation and photon collection with nanometre-scale precision and of long-term stability are challenging. State-of-the-art demonstrations have so far been obtained in laboratory settings with liquid-helium cryostats[4,6] or ultra low-vibration closed-cycle cryocoolers[3,5,7], incorporating bulky optical systems between the source and the collection fibre. A more robust solution for fibre coupling is needed to enable the operation of these bright single-photon sources in standard closed-cycle cryocoolers that are subject to vibration, in the same vein as the development of fibre-coupled superconducting nanowire-based single-photon detectors[23].

In the past fifteen years, there have been many attempts toward this goal[24], exploiting QDs in nanotrumpets[25], photonic crystal cavities[26,27], microlenses[28,29], micromesas[30,31] or bullseye cavities[32]. Most approaches have relied on direct gluing of a fibre on the

---

[1]Quandela SAS, Massy, France. [2]Centre de Nanosciences et de Nanotechnologies, Université Paris-Saclay, CNRS, Palaiseau, France. [3]Quandela GmbH, Munich, Germany. ✉e-mail: niccolo.somaschi@quandela.com

 1

single-photon device, often with limited mode overlap, thereby significantly reducing the efficiency of the fibre-connected source. These studies showed high single-photon purity of the extracted photons while few also demonstrated high photon indistinguishability[33,34], a key metric for most quantum applications. To achieve high photon indistinguishability, the fibre integration must be combined with electrical control of the single-photon source to reduce charge noise effects. So far, the most advanced demonstration was reported in ref. 33, showing a high indistinguishability combined with a brightness at the output fibre ($B_{out-fib}$) of 1.53 %, defined as the probability of getting a single photon at the output fibre of the system for each laser pulse. However, these results were obtained with no deterministic placement of the QDs relative to the cavity mode and no active control of the spectral detuning of the QD line with respect to the cavity resonance. Combining the controlled coupling of the QD to the cavity, the electrical control of the source, and the accurate fibre-pigtailing is highly challenging. Specifically, the fibre alignment with the device should be controlled at the scale of tens of nanometres and be maintained from room temperature to below 10 K, in an architecture which is robust to many cooling cycles.

In the present work, we report how we have tackled and overcome all these technological challenges, opening the door to a new era of quantum photonic applications based on single QD sources. Our technology is based on individual semiconductor QDs in micropillar cavities which are directly coupled to single-mode fibres. We demonstrate the successful integration of our source in a compact two-stage Gifford-McMahon cryocooler operating at a base temperature of 2.4 K. Our mechanical design is shown to precisely maintain the alignment between the cavity and the fibre during cooldown and is robust enough to handle the vibrations of the cryocooler during operation. The optical excitation of the source and the photon collection are

realised through compact optical modules connected to the pigtailed device. The source produces a stream of single photons with an emission rate of 16.7(5) MHz and indistinguishability of 97.5(1) %, and is stable over hours of continuous operation, as well as throughout multiple cooling cycles of the cryocooler.

## Results

### Source design and fibre coupling

Our quantum emitters are annealed self-assembled InGaAs QDs grown by molecular beam epitaxy positioned at the centre of an unbalanced GaAs planar microcavity formed from 20(40)-pairs top (bottom) distributed Bragg reflectors consisting of $\lambda/4$ alternating layers of GaAs and $Al_{0.95}Ga_{0.05}As$. The whole heterostructure is doped with Si (n-doping) for the bottom mirror up to 25 nm below the QD layer and Be (p-doping) for the top mirror. This constitutes a vertical p-i-n junction which is used under reverse-bias to control the charge state of the QD ground state as explained in ref. 35 and to tune the emission wavelength by more than 150 pm through the quantum-confined Stark effect. The self-assembled QDs are randomly distributed in the plane of the wafer and have an inhomogeneous spectral broadening due to variations in size and composition. We use the in-situ lithography technique[36] to precisely locate the QDs on the wafer and fabricate micropillars centred on individual QDs. We choose the diameter of each micropillar to coarsely match the wavelength of the cavity mode to that of the target QD line (within 200 pm). For bias voltage tuning, each micropillar on the chip is connected to electrical contacts by four narrow ridges[37] (Fig. 1a). The number and arrangement of the ridges preserves the Gaussian-like spatial distribution of the fundamental mode. In addition, the electrical contact pads serve as supporting structures for the fibre pigtailing as described later. The resulting micropillars have diameters ranging from 2 to 3.5 $\mu$m and quality

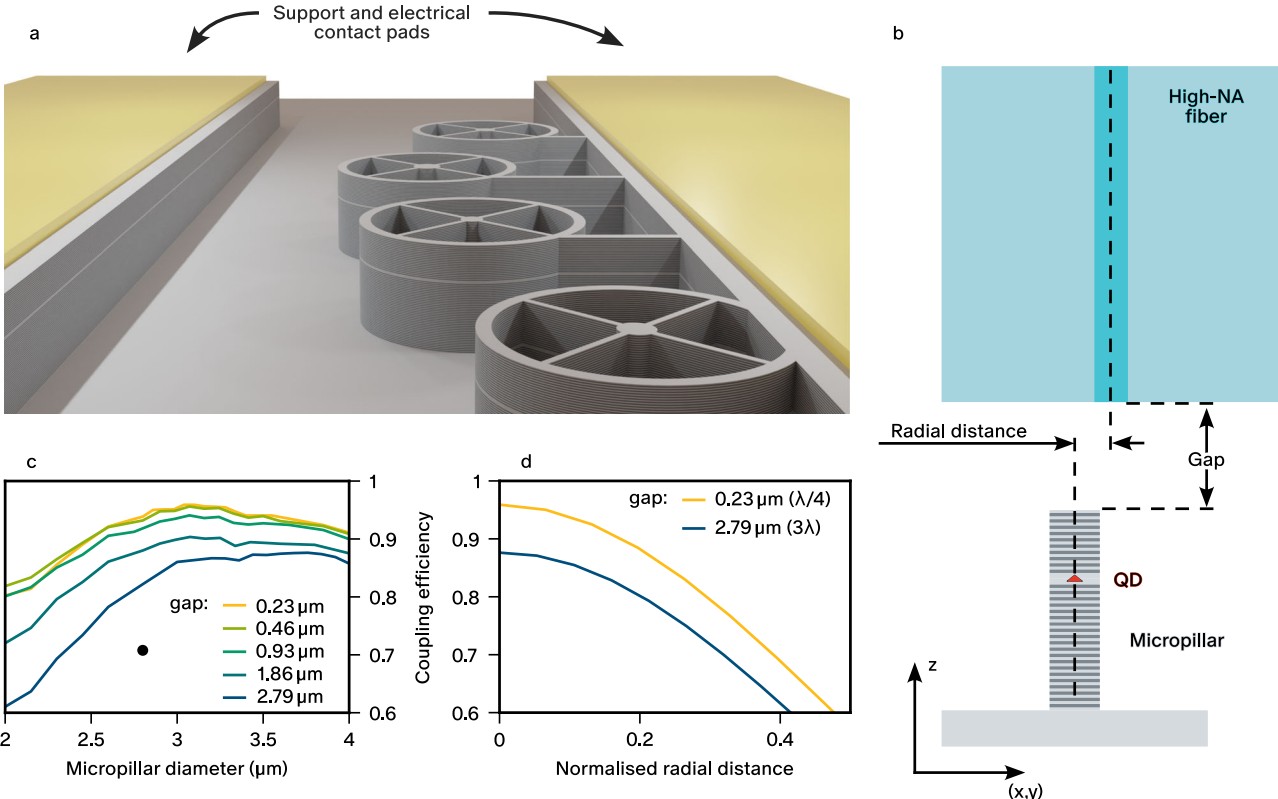

**Fig. 1 | Direct fibre-coupling of micropillar-based single-photon sources. a** 3D render of the deterministic micropillar devices with the electrical contact pads designed to act as mechanical support in the pigtailed device. **b** Illustration of the region of interest around the micropillar and fibre tip, defining important parameters for the alignment of the system. **c**, **d** Numerical simulation of micropillar to high-NA fibre coupling as a function of the gap and of the radial distance normalised to the micropillar radius. The black dot in (**c**) indicates the simulated coupling efficiency for the diameter and gap of our device.

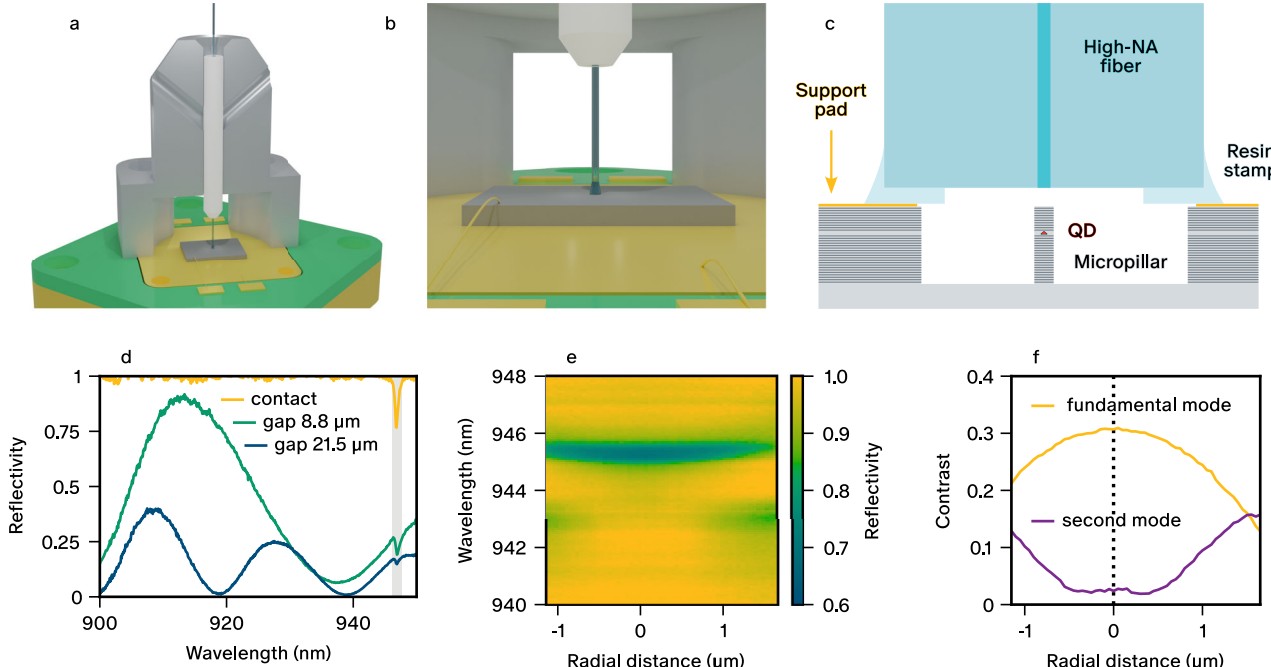

**Fig. 2 | Fibre alignment procedure. a, b** 3D visualisation of the resin stamp and fibre holder (section) for improved stability of the pigtailed system. **c** Sketch of the micropillar-fibre section highlighting the contact physical region between the resin stamp and the single-photon emitting device (not in scale). **d** Reflectivity spectra of the micropillar at room temperature for different vertical gaps showing broadband modulations from interference effects. The fundamental mode spectral region is highlighted. **e** 2D colour map of the reflectivity spectrum at room temperature across the section of a micropillar. **f** Contrast of the two lowest-order modes extracted from (**e**).

factors of around $Q \approx 13{,}000$ which leads to a radiative enhancement of the targeted QD emission lines with Purcell factors $F_p \approx 13$. This results in a decay time $\tau \approx 80$ ps when the QD emission is tuned into resonance with the cavity using the bias voltage control. Chips containing dozens of devices are first characterised at 4 K with a free-space optical setup and the best-performing micropillar in terms of single-photon rate, purity, and indistinguishability is identified for each chip[38]. They are subsequently brought back to room temperature for the direct fibre-coupling of the chosen micropillars.

For an optimal mode matching between the micropillars and single-mode fibres, we choose a commercially available single-mode fibre (UHNA3) with a numerical aperture (NA) of 0.35 and a core diameter of 1.8 $\mu$m. This fibre is spliced to a standard 780HP fibre for connection with the excitation and collection setup. The fibre-micropillar system is sketched in Fig. 1b. The key parameters are the gap that separates the fibre and the micropillar, and the radial distance between their respective optical axes. Figure 1c presents the results of numerical simulations of the coupling efficiency between the ridge-connected micropillar and the high-NA fibre performed with finite-difference time-domain and mode expansion methods. The different colours correspond to different gaps between the fibre and the micropillar. As expected, minimal losses are observed when the gap approaches zero, as there are no losses due to divergence of the beam in the unguided region. We find that for a gap size of 0.23 $\mu$m, the maximum achievable coupling is around 96 %. For a fibre which is perfectly aligned on axis, there is an optimal micropillar diameter which maximises the mode overlap with the fibre. We note that the coupling efficiency around the optimal diameter remains above 94 % over a range of $\pm 0.5\,\mu$m, which is larger than the tolerance in the fabricated diameters. Looking at the curves for the different gap sizes, we estimate the coupling efficiency for the optimal micropillar diameter to decrease to 93.8 % at 1 $\mu$m gap size, and to 89.9 % at 2 $\mu$m gap size. We find that the optimal diameter increases for larger gaps due to smaller divergence of the light emerging from larger micropillar

modes. Finally, Fig. 1d presents the coupling efficiency for various radial displacements between the micropillar and fibre core axes. The results follow the expected convolution between two Gaussian modes evidenced by a slow quadratic variation near zero displacement. This indicates that the presence of ridges does not significantly alter the Gaussian profile of a cylindrical micropillar cavity.

## Fibre pigtailing

Our numerical studies imply that direct contact between the micropillar and the fibre is optimal in terms of coupling efficiency. However, any unwanted source of stress applied by the fibre on the micropillar will lead to wavelength shifts of both the QD emission line and the cavity mode resonance[39,40]. Therefore, we aim for the fibre to be as close as possible to the micropillar without direct contact. This has to be achieved at cryogenic temperatures after thermal contraction of all the components in the mechanical structure, each of them presenting different thermal contraction coefficients. To meet these requirements, we employ the structure presented in Fig. 2a–c. The fibre, previously glued into a ceramic ferrule, is held in place by a titanium fibre-holder assembly. To secure the fibre to the chip, we fabricate a customised supporting structure at the end of the fibre which we refer to as the stamp. The stamp defines a 3 $\mu$m gap by design and allows us to push the fibre on top of the electrical contact pads surrounding the micropillar while guaranteeing that no direct pressure is applied on the micropillar itself. The mechanical stability obtained in this way is needed to prevent sliding of the fibre tip during cooldown and to eliminate relative vibrations between the source and the fibre. The stamp structure is fabricated by dipping the tip of the fibre in uncured resin, and subsequently shaping the resin by pressing the fibre on a Si-based disk-shaped mould. To align the fibre core to the mould, a small gold disk is fabricated at the centre of the mould to allow for spatial alignment by using the reflection of a laser back into the fibre. The resin is then cured by illuminating it with UV light.

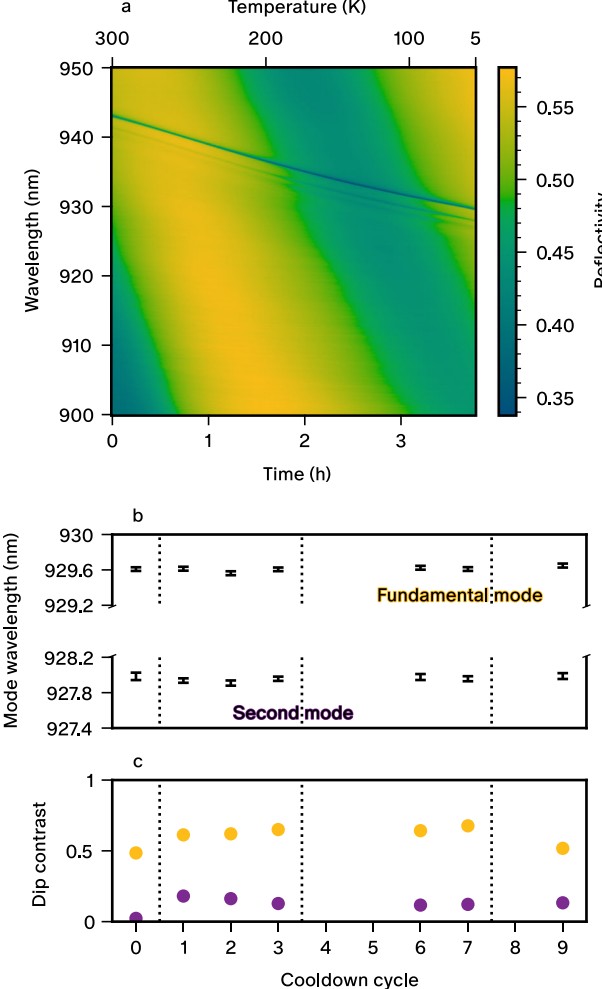

**Fig. 3 | Stability of fibre-pigtailing. a** Reflectivity monitored during a typical cooldown of the pigtailed single-photon source. **b, c** Comparison of reflectivity features (micropillar modes wavelength and contrast) at the cryocooler base temperature in free space and six out of nine successive cooldown cycles in the pigtailed version. The dotted lines indicate a change of cryocooler, which only requires the fibre splice to be redone. Error bars in (**b**) are derived from the instrument calibration and the fitting procedure.

To ensure that the fibre is positioned within a few micrometers above the micropillar, a vertical alignment step is first performed to define the relative position of the ferrule in the fibre holder before securing them together. The ferrule is gradually lowered into the fibre holder and we monitor the reflectivity spectrum resulting from a broadband light source sent through the fibre. A Fabry-Pérot cavity is formed between the fibre end facet and the device surface, leading to a periodic modulation of the reflectivity spectrum. The spectra in Fig. 2d show the reflectivity measured for various vertical distances between the fibre tip and the chip. The data is normalised with respect to when in direct contact. The observed broadband modulation of the reflectivity allows us to extract the gap value as $\frac{\lambda_1 \lambda_2}{2 \Delta \lambda}$, where $\Delta \lambda$ is the spacing between two successive maxima of the reflectivity spectrum at wavelengths $\lambda_1$ and $\lambda_2$. This technique allows us to precisely measure the vertical distance between the fibre facet and the top of the planar microcavity with a relative accuracy which improves linearly while reducing the distance. Hence, the distance is continuously monitored as the fibre is lowered in the holder until it lands smoothly on the contact pads. At this point the ferrule is glued into the fibre-holder.

Once the vertical alignment is done, the fibre holder is separated from the chip and rigidly attached to a precision scanning stage for accurate $x$, $y$, and $z$ positioning, thereby maintaining a gap of a few micrometers between the stamp and the surface of the chip. To position the fibre core precisely above the micropillar centre with sub-$\mu$m lateral alignment precision, we then exploit specific features in the reflectivity spectrum of the micropillar. The micropillars support several higher-order transverse modes with resonance wavelengths shorter than the fundamental cavity mode[41]. These modes are seen as dips in the reflectivity spectrum whose contrasts depend on their spatial overlap with the Gaussian mode of the fibre. Coupling to symmetric (antisymmetric) modes will be maximised (minimised) with perfect centring. The result of such a measurement is presented in Fig. 2e, which displays a reflectivity map as a function of wavelength and radial displacement between the micropillar and the fibre core. We can clearly see the fundamental micropillar mode at 945.8 nm whose contrast is maximised for perfect alignment. The fundamental mode is also highlighted in Fig. 2d with a grey vertical bar. The second transverse mode of the micropillar appears around 943.0 nm with some radial displacement. To visualise the coupling to the different transverse modes more clearly, we plot the contrast of the fundamental and second order modes dips (defined as the normalised depth of the dip with respect to the background) as a function of radial distance. This allows us to accurately align the two structures radially. From the symmetric behaviour of the dip contrast with respect to the lateral displacement, we estimate the alignment precision to be better than 200 nm. Once the $xy$-position is aligned, we close the vertical gap between the stamp and the chip, and we secure the fibre holder to the chip holder using screws. During this final step, the reflectivity spectrum is constantly monitored to check that the transverse alignment is not perturbed. Furthermore, we underline that our fibre-coupling process is fully reversible. The fibre holder can be unscrewed, and another micropillar can be targeted without damage nor alteration to the properties of the micropillars on the chip.

After completion of the direct coupling process, the fibre-pigtailed source is installed in a compact two-stage Gifford-McMahon cryocooler operating at a base temperature of 2.4 K. The fibre-holder assembly is in thermal contact with the cold plate of the cryocooler through the chip holder. To demonstrate the robustness of the alignment, we show the reflectivity spectrum of the pigtailed device as a function of temperature during a typical cooldown in Fig. 3a, starting from 300 K all the way down to the base temperature of the cryocooler. The narrow resonance dips correspond to the micropillar modes that are gradually blue-shifting with decreasing temperature due to the change in refractive index of the cavity medium. More importantly, we do not see the strengthening of higher-order cavity modes at the expense of the fundamental mode, which would indicate a transverse displacement of the fibre with respect to the micropillar (compared to the behaviour observed in Fig. 2e, f). We observe a shift in the slow modulation of the background reflectivity due to a contraction of the vertical gap between the fibre and the micropillar (cf. Fig. 2d). From the background modulation, we estimate the gap to be 3.5 $\mu$m, which is comparable to the design gap size of 3 $\mu$m. Next, we show in Fig. 3b, c that the micropillar modes wavelength and contrast are not significantly altered across nine consecutive thermal cycles in three different cryocoolers over several months of operation. The wavelength fluctuation is less than 30 pm and the second mode contrast is always significantly lower than the fundamental mode one, implying that our technique to hold the fibre achieves remarkable stability with no tangible effect on the microcavity. We note that the first data point is a reference and was taken during the free-space characterisation preceding the pigtailing procedure. The lower contrast in free-space evidences a different mode matching to the fibre using lenses.

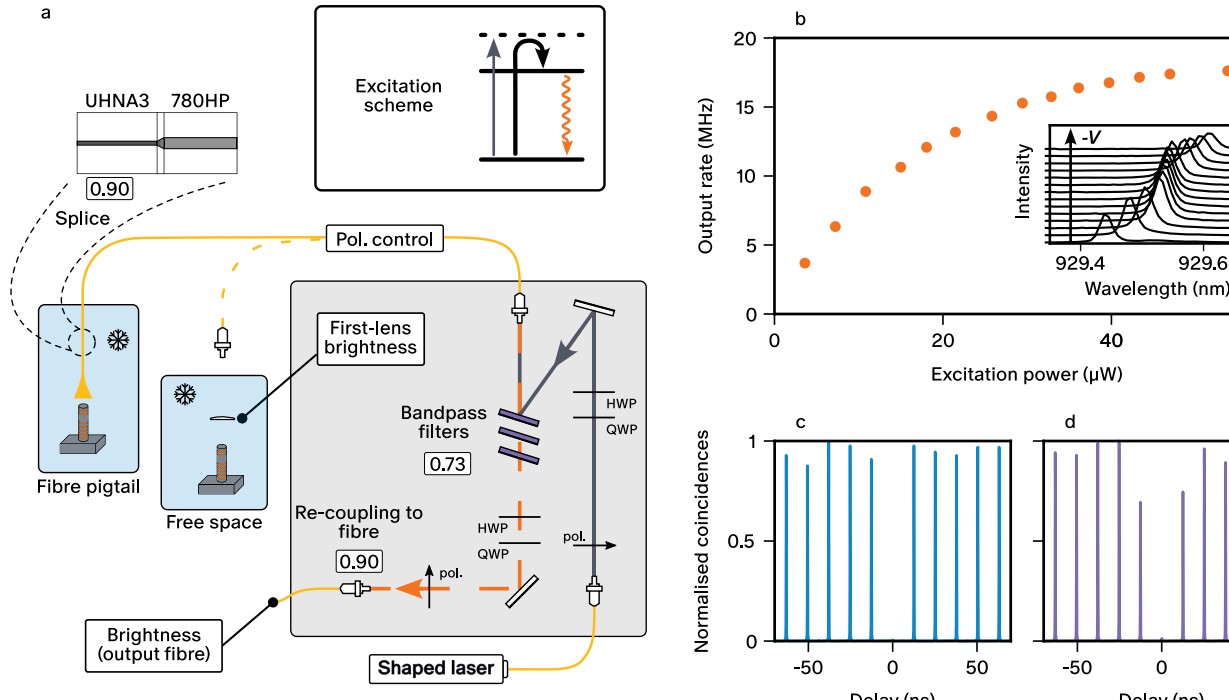

**Fig. 4 | Characterisation of the fibre-coupled single-photon source. a** Schematic of the optical setup used to operate the source. A compact optical setup inserted in a rack system (grey area) allows to send the laser toward the fibre coupled to the QD source and to collect the emitted photons. For the excitation, the laser is reflected onto bandpass filters. For the photon collection, the single photons emitted by the QD are filtered by three bandpass filters and coupled to the system output single mode fibre. Half and quarter wave plates (HWP, QWP) and polarisers are used to control the polarisation after the optical fibres. To quantify the efficiency of the micropillar-to-fibre coupling, the same optical setup is used to measure the source performance in a free space cryostat where the laser is focused on the micropillar with a lens. The numbers indicate the transmission of various parts of the setup. The inset shows the working principle of LA phonon-assisted excitation. **b** Emission rate saturation curve as a function of excitation power. The inset shows the QD emission line as a function of applied voltage with steps of 80 mV. **c** Second-order auto-correlation function and (**d**) Hong-Ou-Mandel interference measurements.

## Performance of the fibre-pigtailed source

We now present the characterisation of the pigtailed source and compare its properties to those measured before fibre integration. Figure 4a presents the schematic of the optical setup used for characterizing the source before and after pigtailing. In both cases, we operate the device with a pulsed near-resonant laser relying on longitudinal acoustic (LA) phonon-assisted excitation[42–45]. In this excitation scheme, represented in the inset of Fig. 4a, the laser wavelength is blue-detuned by less than 1 nm from the QD resonance and is separated from the emitted single photons using spectral filters. This technique is very robust to fluctuations in the laser wavelength and intensity, and allows us to address a single exciton dipole, leading to highly polarised single photon emission[45]. Three narrow bandpass spectral filters (with a transmission of 90 %) with 0.8 nm full width at half maximum are used to block the excitation laser. The filtering stage has a total transmission of 66 % at our operating wavelength.

We coarsely bias the p-i-n junction to control the charge state of the QD and here choose to operate with the neutral exciton state. Such first step allows to minimise any blinking effect that would come from an undefined QD charge state. We then fine-tune the line in resonance with the cavity mode exploiting the quantum-confined Stark effect. An example of fine tuning is presented in the inset of Fig. 4b. An increase of the QD emission intensity is evidenced when the QD is in resonance with the cavity mode. We then optimise the duration of the laser pulses, the laser detuning, and the polarisation in order to maximise the performance of the source. We find the best settings for a pulse duration of 10 ps at a repetition rate of 79.21 MHz and a detuning of −0.8 nm with respect to the QD emission wavelength. Finally, the polarisation of the laser is finely adjusted to excite only one of two orthogonal dipoles of the neutral exciton. The emitted photons are thus highly linearly polarised with the degree of polarisation reaching 95 %.

We measure the single-photon rate at the system output and correct the recorded rate for the efficiency of the detector. Figure 4b represents the output single-photon rate as a function of the time-averaged excitation power. As expected with phonon-assisted excitation, the rate saturates at high excitation power, here at a value of 17.6 MHz. For the best performance in terms of single-photon purity and indistinguishability, we operate the source slightly below saturation, at a single-photon rate of 16.7 MHz, which corresponds to 95 % of the saturation value. This minimises the fraction of spurious laser light collected while maintaining a high single-photon source efficiency and is the regime in which all the further experiments are performed. From the output single-photon rate and the laser repetition rate, we calculate $B_{out-fib} = 20.8(8)$ %.

We use these results to quantify the micropillar-to-fibre coupling efficiency, providing a direct measure of the fibre-pigtailing process quality. To access the losses which are solely due to the imperfect coupling from the micropillar fundamental mode into the fibre, we use results from the characterisation of the source before the fibre-coupling process. A free-space characterisation of the source performances was obtained using a lens to focus the laser on the micropillar and to collect the single photons. The same optical setup for excitation and laser filtering as for the pigtailed source characterisation was used - see Fig. 4a. We estimate the first lens brightness ($B_{fl}$) of our device, defined as the probability of collecting a photon per excitation pulse at the output of the micropillar in a 0.7 NA lens to be $B_{fl} = 46.8(2.5)$ %. To obtain this value, the coupling of the photons to the fibre through the lens was independently measured by exploiting the birefringence of the micropillar cavity[46]. Assuming that $B_{fl}$ is unaltered by the pigtailing

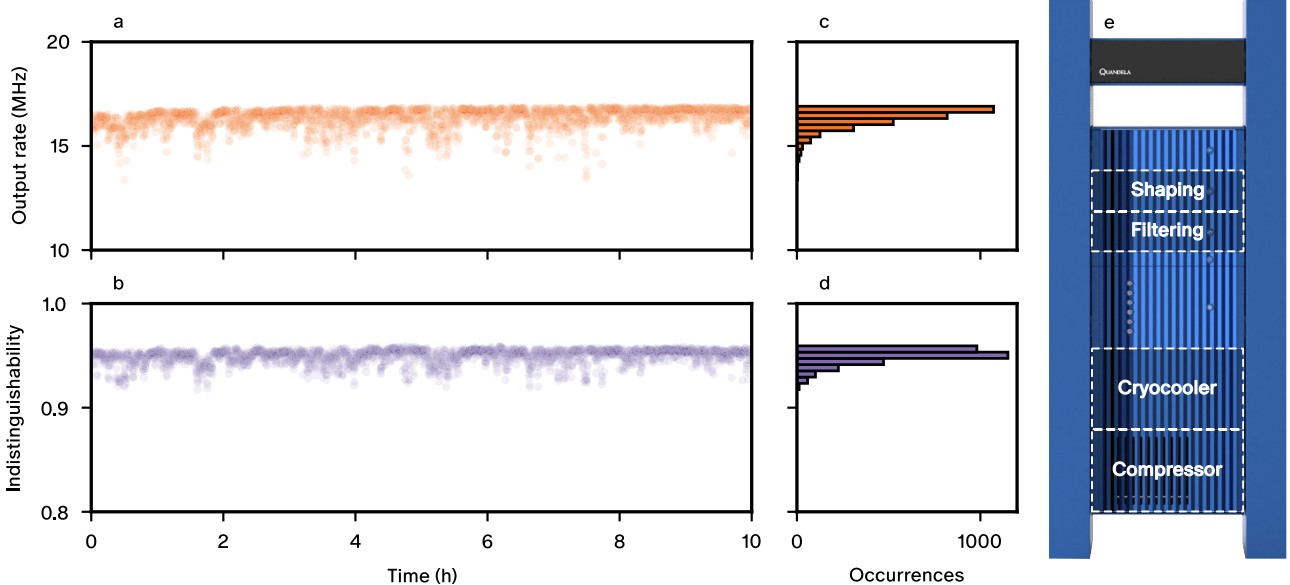

**Fig. 5 | Stability performance. (a,b)** Stability over ten hours of the single-photon rate at the fibre output and photon indistinguishability and **(c,d)** corresponding histograms. **(e)** Standalone single-photon source system with interconnected modules dedicated to cryogenics, excitation and collection.

## Table 1 | Comparison of the best fibre-pigtailed sources based on semiconductor QDs

| ref. | Excitation scheme | $B_{coll-fib}$ (%) | $B_{out-fib}$ (%) | $g^{(2)}(0)$ (%) | $M$ (%) |
|---|---|---|---|---|---|
| 25 | non resonant | 5.8 | 0.35 | - | - |
| 27 | p-shell | 10.9 | 5.74 | 20 | - |
| 28 | non resonant | 0.28 | 0.00085 | 7 | - |
| 33 | resonant | 5 | 1.53 | 3.7 | 90 |
| 31 | non resonant | - | 0.034 | 15 | - |
| 32 | non resonant | 8.1 | 1.6 | 16 | - |
| 34 | p-shell | 13.4 | 0.44 | 0.7 | 82 |
| This work | LA | 35.1 | 20.8 | 1.26 | 97.5 |
| 7 | resonant | 80.6 | 71.2 | 2.05 | 98.56 |

The last line of the table reports on the performance of the best single-photon source reported so far based on a QD inserted in a open cavity platform operating in a ultra-low vibration cryocooler[7].

process, and considering that the transmission of the fibre splice between high-NA and standard fibres is measured to be 90(2) %, and the transmission of the filtering system (66(2) %), we infer a micropillar-to-fibre coupling of 75(5) %. The detailed loss budget of the setup is provided in Table 2 of the methods section. The measured coupling efficiency is comparable within error bars to the expected value of 71 % predicted by numerical simulations for our gap (3.5 μm) and micropillar size (2.8 μm).

Next, we investigate the quality of the source by measuring the single-photon purity and indistinguishability of the emitted photons. In Fig. 4c we show the pulsed second-order intensity autocorrelation function, exhibiting strong photon antibunching at zero time delay with $g^{(2)}(0) = 1.3(1)$ %. In Fig. 4d we also show the outcome of Hong-Ou-Mandel (HOM) interferometry which gives a two-photon HOM visibility of $V_{HOM} = 95.0(1)$ %. After correction for the multi-photon events and the small imperfections of the interferometer[47], we obtain an indistinguishability $M = 97.5(1)$ %. Such value is on par with the highest values reported for QD-micropillar cavities[3] considering that the rather broad (0.8 nm) spectral filtering hardly suppresses the phonon side-bands contributions. This is confirmed by the observation of an increased indistinguishability from $M = 94.3(1)$ % in the free-space characterisation setup, which we attribute to the lower working temperature of 2.4 K for the cryocooler hosting the fibre-coupled device compared to 5 K for the low-vibration cryocooler used for the free-space characterisation. While strongly suppressed by the acceleration of the zero-phonon line (see refs. 48,49), the residual contribution of the phonon sidebands is further reduced at lower temperatures.

Once the source is set to its best working conditions, we test its stability over ten hours of continuous operation. In this measurement, only the excitation laser power is actively stabilised, with no stabilisation of the fibre polarisation nor of the bias voltage. In Fig. 5a,b we show the corresponding time traces of the single-photon rate and the photon indistinguishability. Their distribution is depicted in Fig. 5c, d, characterised by a relative standard deviation of 2.82 % for the single-photon rate and of 0.69 % for the indistinguishability. We estimate that a variation of single photon rate of less than 3 % would correspond to a relative displacement of the fibre-micropillar alignment by 100 nm. We thus attribute the observed deviations from the maximum value mainly to instabilities in the electrical environment of the QD which slightly modifies the QD-cavity detuning hence the indistinguishability. Further work is needed to understand the influence of various device parameters on the electrical noise: the nature of the dopants, the overall QD density, or the distance to doping layers.

## Discussion

In conclusion, we have demonstrated a reliable and efficient method for the pigtailing of micropillar-based single-photon sources that enables easy integration into compact cryocoolers. We have described the fibre-coupling process and its main challenges, and presented the results of reflectivity measurements to check the coupling stability over time. Our pigtailed single-photon source shows record performances in brightness at the output fibre (20.8 %) and indistinguishability (97.5 %). To put the present results in perspective with respect to the state of the art, the performances of the best fibre-pigtailed sources are given in Table 1. The highest brightness at the output fibre of a pigtailed single-photon source before the present work was $B_{out-fib} = 5.74 $%[27] for a $g^{(2)}(0)$ of around 20 % and no indistinguishability measurement. In our case, this four-time increased $B_{out-fib}$ is reached for an indistinguishability of 97.5 % when the highest value so far was 90 % for a brightness at the output fibre of only 1.53 %[33]. This comparison highlights that our results define a new state-of-the art for QD-based

fibre-pigtailed single-photon sources. This is achieved in a fully deterministic device, including the QD controllably coupled spatially and spectrally to a confined cavity mode, and the fibre accurately aligned on said cavity mode. Moreover, the performances are shown to be stable over hours of uninterrupted operation and the coupling is maintained intact for months in different cryocoolers. We also include in the last line of Table 1 the current state of the art of QD-based single photon source (with record brightness of $B_{out-fib} = 71.2\%$), achieved with an open cavity system for a QD operated in an ultra-low vibration cryocooler and free space optics. Such high value illustrates the potential for improvement and a near-term target for plug-and-play, compact and integrated systems.

To further improve the efficiency of our system there are several parallel approaches that can be followed. Numerical simulations show that the outcoupling efficiency $\eta_{top}$ of our micropillar is currently limited by a deviation from the ideal verticality by one or two degrees, leading to the current value of $\eta_{top} \approx 65\%$, when $\eta_{top} = 93\%$ is achievable. The use of near-resonant excitation, while providing strong stability results in an occupation probability ($p_{occ}$) of the QD target state estimated to be 70% for the detuning of −0.8 nm used here. Switching to the resonant excitation scheme would allow $p_{occ} = 98\%$ as measured in ref. 50, which could be combined with a polarised Purcell effect as in ref. 7 to push the probability to emit a photon in a defined mode of the micropillar around $p_{occ}\beta \approx 90\%$. Regarding the pigtailing process, it is possible to engineer the resin stamp on the fibre tip to approach it even closer and operate the system in contact with the micropillar to reach $\eta_{fibre-coupling} = 96\%$. An advanced fibre splicing should allow reaching $\eta_{splice} = 95\%$ with optical setup for resonant excitation polarisation filtering now approaching efficiency around $\eta_{setup} = 90\%$. All these improvements combined should allow reaching the threshold of 2/3 for measurement-based quantum computation[51] considering the detector efficiencies currently above 98%[52].

It is worth noting that the presented fibre-pigtailing method avoids contact with the source while proving stable assembly over extended periods of time and cooling cycles. This feature makes our approach particularly suited for fragile photon-extraction structures. The present pigtailing technique is thus compatible with different cavity designs with emission out of plane such as bullseyes, nano-trumpets, and nanowires among others, which have all allowed to obtain efficient single photon collection[53–55].

The robust fibre pigtailing solution presented in this work demonstrates great potential for applications of single-photon sources beyond academic laboratories. All the components needed for operating our single-photon sources are mounted in a rack system as depicted in Fig. 5e. This includes the compressor and the cryocooler hosting the pigtailed single-photon source, the pulsed laser with its spectral shaping module, the optics for filtering the single-photon emission, and a module for single-photon purity and indistinguishability measurements. Moreover, it is possible to host multiple sources within the same cryocooler, which reduces the energy requirements compared to running a separate cryocooler for each source. We estimate a total power consumption of 3 kW during operation. The stability of the system can be further improved by implementing an active stabilisation of the fibre polarisation and an optimisation of the bias voltage, especially for longer operation periods[56]. As such, the single-photon source can be operated as a genuine turn-key system.

## Methods
### Numerical simulations
We use the Meep software package to perform three-dimensional finite-difference time-domain electromagnetic simulations of the fibre-coupled micropillar devices[57]. A broadband, linearly-polarised dipole source is placed in the middle of the cavity to excite the dielectric structure. We record the local density of states spectrum of the dipole and the Poynting flux spectra through the six faces of the

**Table 2 | Summary of the measured and calculated efficiencies of the free space and fibre-pigtailed systems**

|  | Free space | Fibre-pigtailed |
| --- | --- | --- |
| Detected countrate (MHz) | 5.80 ± 0.05 | 4.78 ± 0.05 |
| APD quantum efficiency | 0.28 ± 0.01 | 0.29 ± 0.01 |
| Laser repetition rate (MHz) | 79.42 ± 0.01 | 79.21 ± 0.01 |
| Brightness (output fibre) | 0.261 ± 0.010* | 0.208 ± 0.008* |
| Filtering setup transmission | 0.68 ± 0.02 | 0.66 ± 0.02 |
| High-NA to 780HP splice | – | 0.90 ± 0.02 |
| Micropillar-fibre coupling | 0.82 ± 0.02 | 0.75 ± 0.05* |
| Brightness (first lens) | 0.468 ± 0.025* | |

(*calculated value).

rectangular simulation space[58]. In addition, we use a mode decomposition monitor for the simulation's top plane to extract the flux in the fundamental mode of the high-NA fibre. The simulation is ended once the fields have sufficiently decayed, by monitoring the total electromagnetic energy in the simulation volume. The micropillar to fibre coupling is computed by dividing the fundamental mode flux spectra with the top flux spectra. To speed-up the simulations, the number of top and bottom pairs are reduced to obtain a similar cavity quality factor than in the experimental case. We keep the same relative contribution to the quality factor of the top and bottom DBRs.

### Operation of the single-photon source
The QD-based single-photon source is operated with pulsed optical excitation. A femtosecond laser with a repetition rate of 79.21 MHz is spectrally shaped with a slit positioned in the Fourier plane of a 4f optical system to carve out 10 ps long pulses with adjustable central wavelength. The detuning of the excitation is adjusted to be 0.8 nm bluer than the QD emission line to excite the emitter through the longitudinal acoustic phonon-assisted scheme. Spurious laser light collected with the emitted single photons is spectrally filtered with a set of three 0.8 nm-wide bandpass filters. The single correlations measurements were performed with Single Quantum superconducting nanowires single-photon detectors (detection efficiency 90%, timing jitter 20 ps). The emission rate saturation and the stability performance were measured with Excelitas single-photon avalanche diodes (maximum detection efficiency 31%, dead time 24 ns). Both were used in combination with a Swabian Instruments time tagger.

To extract the indistinguishability ($M$) from the Hong-Ou-Mandel visibility ($V_{HOM}$) the following formula from ref. 47 was used: $M = \frac{V_{HOM} + 4RT[1 + g^{(2)}(0)] - 1}{4RT[1 - g^{(2)}(0)]}$, where $R$ ($T$) is the imperfect reflection (transmission) of the second beam splitter of the Hong-Ou-Mandel interferometer.

### Loss budget and rates
The measured rates, transmission of optical elements, and calculated performance of the system are summarised in Table 2. For both the free space and fibre-pigtailed operation of the device, the total efficiency is obtained from the measured countrate and the laser repetition rate.

## Data availability
The source data used in this study are available in the Figshare database[59].

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

## Acknowledgements
This work has been partly funded by the European Commission as part of the EIC accelerator program under the grant agreement 190188855 for the SEPOQC project, by the Horizon-CL4 program under the grant agreement 101135288 for the EPIQUE project, by the European Union's Horizon 2020 Research and Innovation Programme QUDOT-TECH under the Marie Skłodowska-Curie grant agreement 861097, by the Plan France 2030 through the projects ANR-22-PETQ-0011 and ANR-22-PETQ-0013, and by the French RENATECH network.

## Author contributions
F.P. developed the fibre-pigtailing technique with inputs from V.G., N.S., P. Senellart, S.B., and T.H.A.; F.P. implemented the pigtailing together with T.B.; N. Margaria, P. Stepanov, and A.F.A. conducted most of the experimental investigation and data analysis with support from M.P. and M.D.; M.B. and E.I. integrated the fibre-pigtailed source in the cryocooler and provided the optical modules for plug-and-play operation; R.S. and S.B. designed and performed the numerical simulations; W.H., A.B., A.P., M.M., A.L., S.B., T.H.A. performed the design, growth, and fabrication of the single-photon source. All authors participated to scientific discussions; T.H.A., S.B., N. Maring, and P. Senellart supervised the project; N. Margaria, W.H., A.F.A., M.D., T.V., S.B., T.H.A., and P. Senellart wrote the paper with feedback from all authors.

## Competing interests
N.S. and P. Senellart are co-founders of the company Quandela. A patent (WO/2022/229559) on the fibre-pigtailing method has been filed listing F.P., N.S., and V.G. as inventors. The remaining authors declare no competing interests.
