## [Transparent Peer Review file · Nature Communications]

Efficient fibre-pigtailed source of indistinguishable single photons

Corresponding Author: Dr Nico Margaria

Version 0:

Reviewer comments:

Reviewer #1

(Remarks to the Author)

The manuscript presents an advancement in the development of an efficient fiber-pigtailed single-photon source based on semiconductor quantum dots, and the reported results are promising. However, the system is not yet a plug-and-play solution, as it still requires the use of additional filter systems. Furthermore, a more thorough comparison with existing literature is necessary to properly contextualize these findings within the current state of the field. Below are my detailed comments:

1) The authors utilize single-photon avalanche diodes (APDs) for photon detection, adjusting for the detector's efficiency. While this is standard practice, they should provide more details on the use of filters to block the excitation light. A clearer explanation of how these filters contribute to the system's efficiency and the resulting purity of the single-photon emission would offer a more complete understanding of the experimental setup.

2) The manuscript attributes charge noise to the use of beryllium in the p-i-n junction. Given the well-known blinking behavior of quantum dots, which can significantly impact charge noise, it would be beneficial if the authors could provide additional data on the presence or absence of blinking in their quantum dots. This information would be valuable for evaluating the full range of factors contributing to charge noise.

3) Although the inclusion of a p-i-n structure is mentioned, its role in the system's performance is not clearly explained. It would be useful for the authors to present data or an analysis that clarifies how the p-i-n structure influences the charge state or wavelength tuning of the quantum dots. This would help to better understand the structural contributions to the overall performance of the device.

4) The manuscript mentions that single-photon purity and indistinguishability were measured at the system output, but it is unclear whether these measurements were made at the saturation point of the excitation power. Providing g^2 and Hong-Ou-Mandel (HOM) visibility data across a range of excitation powers would help assess how these key parameters vary with excitation conditions.

5) The authors introduce a novel approach to fiber-pigtailed single-photon sources, achieving high-performance metrics. However, the manuscript would benefit from a direct comparison with other fiber-coupled systems and the current state-of-the-art single-photon sources. A comparative analysis would place the authors' work within the broader scientific context, highlighting both the innovations in their approach and potential avenues for further improvement.

Reviewer #2

(Remarks to the Author)

GaAs-based quantum dots in microcavities are the basis for high-quality quantum light sources, and robust single-mode fiber coupling is essential for use in photonic quantum technologies but also for broader use in physics. The authors show such a source based on a novel fiber coupling design.

The authors start with a complete overview of state of the art, and I agree that none of previously demonstrated approaches simultaneously fulfils all requirements (robustness to multiple cooldowns, purity, indistinguishability, and high in-fiber brightness). A recent manuscript gets closest [33] but the real in-fiber brightness is not very clear and indistinguishability is

significantly lower.

Here, the authors achieve a robust cavity-fiber coupling efficiency of 75±5% with a theoretical maximum of 96%, resulting in the end (mainly imperfect QD excitation) of 20.8% in-fiber brightness, with excellent g_2 HBT of 1.3% and HOM wavefunction overlap of $M=0.975$. This is possible by a novel design with a small gap between the fiber and the micropillar, which avoids stress on the micropillar and associated uncontrolled wavelength shifts of the cavity and/or quantum dot. This is done in a process where a ring-shaped resin is applied to the fiber tip using a special stamp, thereafter this ring is secured to the device contact pads using a stable mount.

The work shows a milestone progress for quantum photonics. Although if the achieved brightness is probably not yet sufficient for real-world photonic quantum computers, it shows that careful engineering will lead us there. I find that the contact-free method has significant novelty and originality, and the detailed description and characterization make reproduction as feasible as possible. The manuscript I find excellent in all necessary dimensions, I only have a few questions and comments:

* The authors have included a short outlook section, but the short-term achievable improvements are not very clear to me - can such a single photon source beat some MBQC computing threshold, including or ignoring de-multiplexing?

* What other interesting aspects might help to reach close-to-unity brightness? The authors mention better adjusted micropillar diameter, but is there not always a wavefront curvature mismatch that requires a lens or small piece of graded-index fiber spliced to the end?

* The authors mention that another excitation technique is needed to increase M significantly - do they have data on achievable through-fiber cross-polarization contrast ratios?

* About the brightness value: "From the output single-photon rate and the laser repetition rate, we calculate a fibred brightness of $20.8 \pm 0.8 \%$ ". How does this compare to the other mentioned numbers, it is yet not fully clear to me? Is the brightness measured in the 780HP fiber, is filtering taken into account? In the efficiency budget below, I still get 0.31 and not around 0.2:

0.7 excitation probability
0.75 cav-fiber coupl
0.66 filtering
0.9 splice
=0.31

* I first was wondering why the cryocoolers have been exchanged twice (Fig. 3) - has the device really been completely remounted into different cryocooler cryostats?

* Why has the gap been chosen to be as big as 3.5 μ m, leading to reduced coupling?

Reviewer #3

(Remarks to the Author)

A quantum dot (QD) in a micropillar is a well developed single photon source (SPS). The present paper describes a technique to create a fiber-pigtailed SPS. The SPS has excellent g_2 and HOM metrics. The efficiency ("fibred brightness") is 20.8%. This efficiency is much higher than previous fiber-pigtailed SPSs but considerably lower than state-of-the-art SPSs.

The fiber-pigtailling is carried out by positioning a fiber using the cavity modes as a guide. The fiber is locked in place at room temperature and the positioning is maintained on cooling down. This is nice I think.

Important comments:

1. The paper focusses exclusively on engineering of micropillars. SPS is a broad field, but the engineering in the paper has a narrow scope. Are there wider consequences of the work?

2. The long-term stability Fig. 5 is good but Fig. 5 probes only noise at very low frequencies. The cryocooler produces mechanical noise (maybe also electrical noise) at specific frequencies. How does the SPS respond at these specific frequencies? For instance, is there any change in flux at the main ~ 1 Hz frequency of the cryocooler? The authors mention "to eliminate vibrations" - what's the evidence that this is achieved?

3. HOM - results are impressive but are reported only on interfering successive photons, i.e., photons created 12 ns apart in time. For any multi-photon applications involving demultiplexing the HOM needs to be maintained for photons created far apart in time. What can the authors say about this? And HOM is still less than Ref. 3 - why?

Minor comments:

1. Why use "brightness" for 20.8%? I would suggest that 20.8% is better described as an overall efficiency or something like that. What does "brightness" mean in this context? I would say that it's efficiency x repetition rate.
2. does fiber act as a waveplate - does a linear polarization at input give a linear polarization at output?

3. "fibred brightness" implies that there is a verb "to fiber" yet "fiber" is a noun.
4. $F_p=13$ implies a beta-factor $F_p/(F_p+1)=93\%$, setting (I think) the maximum first-lens efficiency. What are the loss mechanisms leading to 46.8%?
5. why splice the high-NA fiber to a regular fiber? This introduces a mode-mismatch and loss.

Version 1:

Reviewer comments:

Reviewer #1

(Remarks to the Author)

The authors have addressed all the issues I presented, the manuscript is much improved, and ready to be published I believe.

Reviewer #2

(Remarks to the Author)

The authors have carefully addressed all comments and added useful additional information to the manuscript. I have no further comments and recommend publication.

Reviewer #3

(Remarks to the Author)

The authors have responded well to my comments and I recommend publication in Nature Communications.

I have only one final comment, possibly pedantic. I question the use of the term "first lens brightness". The whole point of the paper is that the "first lens" is avoided - it's replaced with the high-NA fiber. The authors should consider this but they can decide as they like.

Response to reviewers

We thank all three reviewers for providing valuable feedback on our manuscript. All three reviewers are generally positive about our work. Reviewer 2 explicitly acknowledges that the present work constitutes a “milestone progress for quantum photonics” and finds the manuscript excellent in all necessary dimensions. We agree with the milestone statement but acknowledge at the same time that some technical aspects in the manuscript could have been presented in a clearer fashion. We agree specifically that a direct comparison with current state-of-the-art single-photon sources in a graph or table is useful, and we have added this important aspect into the manuscript. This then directly highlights the milestone achievement our work constitutes. In the following we address in more detail and individually the issues/questions raised by the respective reviewers.

Reviewer #1 (Remarks to the Author):

The manuscript presents an advancement in the development of an efficient fiber-pigtailed single-photon source based on semiconductor quantum dots, and the reported results are promising. However, the system is not yet a plug-and-play solution, as it still requires the use of additional filter systems. Furthermore, a more thorough comparison with existing literature is necessary to properly contextualize these findings within the current state of the field. Below are my detailed comments:

1) The authors utilize single-photon avalanche diodes (APDs) for photon detection, adjusting for the detector's efficiency. While this is standard practice, they should provide more details on the use of filters to block the excitation light. A clearer explanation of how these filters contribute to the system's efficiency and the resulting purity of the single-photon emission would offer a more complete understanding of the experimental setup.

Answer: Before giving a detailed account of the filters and their performance, we would like to highlight that the reported single-photon source, as delivered to customers (see figure 5e), does indeed constitute a plug-and-play solution. All the parts, including excitation source, filters, etc. are included in the rack system, and the photons are readily delivered through a single output fibre, with no additional filters required on the customer side.

The filtering stage includes three narrow bandpass spectral filters (0.8 nm FWHM, custom Alluxa filters) which block the excitation laser that is less than 1 nm away from the QD resonance. The use of three filters is necessary to obtain a good enough single photon purity at such small detuning. The filtering stage has a total transmission of 66 % at our operating wavelength. The single photons pass through the filters with a transmission of 90 % for each filter, resulting in a total transmission of 73 %, which is further reduced to 66 % by considering also the imperfect collimation from and to the fibre, and the losses through other optical elements used for polarisation control.

Changes in the text: Following the reviewer's comments, we have added a detailed explanation of the system components and their functionality in the main text, including the filtering stage in the new fig. 4a.

2) The manuscript attributes charge noise to the use of beryllium in the p-i-n junction. Given the well-known blinking behavior of quantum dots, which can significantly impact charge noise, it would be beneficial if the authors could provide additional data on the presence or absence of blinking in their quantum dots. This information would be valuable for evaluating the full range of factors contributing to charge noise.

Answer: We thank the reviewer for this question. Indeed, we must correct the statement we made about the influence of beryllium doping on the charge noise experienced by our source.

Concerning the QD blinking: as now better explained in the main text, we first coarsely tune the voltage source to control the charge state. Here we have set this voltage to ensure a stable occupation of the QD in a neutral state (no charge in the ground state). As such, the blinking is minimal in the current measurements. However, we have since conducted a detailed spectroscopic study of the noise in our gated micropillar devices. We are planning to publish a dedicated manuscript on the topic, comparing different potential noise sources such as for example different dopants (Be vs C), overall quantum dot density or distance to doping layers.

Changes in the text:

We coarsely bias the p-i-n junction to control the charge state of the QD and here choose to operate with the neutral exciton state. Such first step allows to minimize any blinking effect that would come from an undefined QD charge state. [...]

We attribute the observed deviations from the maximum value mainly to instabilities in the electrical environment of the QD [...]. Further work is needed to understand the influence of various device parameters on the electrical noise: the nature of the dopants, the overall QD density, or the distance to doping layers.

3) Although the inclusion of a p-i-n structure is mentioned, its role in the system's performance is not clearly explained. It would be useful for the authors to present data or an analysis that clarifies how the p-i-n structure influences the charge state or wavelength tuning of the quantum dots. This would help to better understand the structural contributions to the overall performance of the device.

Answer: This question is directly related to the previous question. Our device includes a p-i-n junction with a doping structure distributed along all the DBR layers: n doping in the bottom mirror up to 25 nm below the QD layer, p-doping in the top mirror. Such structure allows first to control the charge state in the QD by allowing a charge (electron or hole) or none to tunnel from the contact layers to the QD – see for instance ref. [35]. We thus chose a voltage that allows for a stable charge state. The same voltage allows then, within a small voltage range, to fine tune the chosen QD charge state to be in resonance with the cavity mode.

We have tried to make this explanation clearer in the main text, and we have added experimental data in figure 4b (insert) showing the fine spectral tuning of the QD in resonance with the cavity.

Changes in the text:

Our quantum emitters are annealed self-assembled InGaAs QDs grown by molecular beam epitaxy positioned at the centre of an unbalanced GaAs planar microcavity formed from 20(40)-pairs top(bottom) distributed Bragg reflectors consisting of $\lambda/4$ alternating layers of GaAs and Al_{0.95}Ga_{0.05}As.

The whole heterostructure is doped with Si (n-doping) for the bottom mirror up to 25 nm below the QD layer and Be (p-doping) for the top mirror. This constitutes a vertical p-i-n junction which is used under reverse-biased to control the charge state of the QD ground state as explained in ref. [35] and to tune the emission wavelength by more than 150 pm through the quantum-confined Stark effect.

4) The manuscript mentions that single-photon purity and indistinguishability were measured at the system output, but it is unclear whether these measurements were made at the saturation point of the excitation power. Providing g^2 and Hong-Ou-Mandel (HOM) visibility data across a range of excitation powers would help assess how these key parameters vary with excitation conditions.

Answer: All the g_2 and HOM measurements presented in the manuscript were performed with the source operating at an excitation power of 30 μ W, which (according to Figure 4b) produces a photon flux corresponding to 95 % of the maximal photon flux reached at full saturation.

The saturation curve presented in figure 4b represents only the photon countrate. We did not take the corresponding g_2 and HOM visibility data for all the excitation powers. In the experiment, however, we did

observe a substantial degradation of the two single-photon quality measures for excitation powers beyond $30 \mu\text{W}$.

Changes in the text:

For the best performance in terms of single-photon purity and indistinguishability, we operate the source slightly below saturation, at a single-photon rate of 16.7 MHz, which corresponds to 95 % of the saturation value.

This minimizes the fraction of spurious laser light collected while maintaining a high single-photon source efficiency and is the regime in which all the further experiments are performed.

5) The authors introduce a novel approach to fiber-pigtailing for single-photon sources, achieving high-performance metrics. However, the manuscript would benefit from a direct comparison with other fiber-coupled systems and the current state-of-the-art single-photon sources. A comparative analysis would place the authors' work within the broader scientific context, highlighting both the innovations in their approach and potential avenues for further improvement.

Answer: We have now included an in-depth comparison of the performance metrics for the best fibre-pigtailed and free-space sources to date in the discussion section. This clearly highlights the strength of our current results.

The added table particularly shows the improvement compared to previously published performance metrics of fibre-pigtailed sources. It also contrasts our numbers with the best overall single-photon sources based on QDs in open cavities – operated in a laboratory with ultra-low vibration cryocooler, with mostly free space optics. Such comparison highlights that our current numbers are not too far away except for the brightness parameter which is the key figure of merit to improve in the future.

Changes in the text:

Our pigtailed single-photon source shows record performances in brightness at the output fibre (20.8 %) and indistinguishability (97.5 %). To put the present results in perspective with respect to the state of the art, the performances of the best fibre-pigtailed sources are given in Table I. The highest brightness at the output fibre of a pigtailed single-photon source before the present work was $B_{\text{out-fib}} = 5.74 \%$ [27] for a $g^{(2)}(0)$ of around 20 % and no indistinguishability measurement. In our case, this four-time increased brightness is reached for an indistinguishability of 97.5 % when the highest value so far was 90 % for a brightness at the output fibre of only 1.53 % [33]. This comparison highlights that our results define a new state-of-the art in terms QD-based fibre-pigtailed single photon sources. [...]

We also include in the last line of Table 1 the current state of the art of QD-based single photon sources (with record brightness $B_{\text{out-fib}} = 71.2 \%$), achieved with an open cavity system for a QD operated in an ultra-low vibration cryocooler and free space optics. Such high value illustrates the potential for improvement and a near-term target for plug-and-play, compact, and integrated systems.

Reviewer #2 (Remarks to the Author):

GaAs-based quantum dots in microcavities are the basis for high-quality quantum light sources, and robust single-mode fiber coupling is essential for use in photonic quantum technologies but also for broader use in physics. The authors show such a source based on a novel fiber coupling design. The authors start with a complete overview of state of the art, and I agree that none of previously demonstrated approaches simultaneously fulfils all requirements (robustness to multiple cooldowns, purity, indistinguishability, and high in-fiber brightness). A recent manuscript gets closest [33] but the real in-fiber brightness is not very clear and indistinguishability is significantly lower.

Here, the authors achieve a robust cavity-fiber coupling efficiency of 75+-5% with a theoretical maximum of 96%, resulting in the end (mainly imperfect QD excitation) of 20.8% in-fiber brightness, with excellent g_2 -HBT of 1.3% and HOM wavefunction overlap of $M=0.975$. This is possible by a novel design with a small gap between the fiber and the micropillar, which avoids stress on the micropillar and associated uncontrolled wavelength shifts of the cavity and/or quantum dot. This is done in a process where a ring-shaped resin is applied to the fiber tip using a special stamp, thereafter this ring is secured to the device contact pads using a stable mount. The work shows a milestone progress for quantum photonics. Although if the achieved brightness is probably not yet sufficient for real-world photonic quantum computers, it shows that careful engineering will lead us there. I find that the contact-free method has significant novelty and originality, and the detailed description and characterization make reproduction as feasible as possible. The manuscript I find excellent in all necessary dimensions, I only have a few questions and comments:

Answer: We thank the reviewer for this very positive assessment of our work and for the comments that we addressed below to further improve our manuscript.

1. The authors have included a short outlook section, but the short-term achievable improvements are not very clear to me - can such a single photon source beat some MBQC computing threshold, including or ignoring de-multiplexing?
2. What other interesting aspects might help to reach close-to-unity brightness? The authors mention better adjusted micropillar diameter, but is there not always a wavefront curvature mismatch that requires a lens or small piece of graded-index fiber spliced to the end?

*Answer: We thank the reviewer for these two comments that we address together below. We can indeed consider for instance the MBQC threshold given by T. Rudolph and co-workers in their original paper [Phys. Rev. Lett. 100, 060502 (2008)] $\eta_{\text{source}} * \eta_{\text{detector}} = 2/3$. Considering the state of the art in detector efficiency above 98 %, this means that an output fibre brightness of $B_{\text{out-fib}} = 70$ % would allow reaching this threshold.*

Let's detail the various improvement that would allow to reach such value:

Improvements on the source itself:

- We conducted numerical studies showing that the outcoupling efficiency η_{top} of our micropillar cavities are currently limited by the non-perfect verticality of the etching process. Our simulations show that for

the current etching angle of around 2° , the maximal outcoupling efficiency is around 65 %. This is the main limitation of our source. Reducing this angle to 1° would push this value to 85 % and to 0° to $\eta_{\text{top}} = 93$ %.

- In the current work, we use a near-resonant phonon assisted excitation scheme that provides a great stability but leads to an occupation probability (p_{occ}) estimated to be 70 % at 0.8 nm laser detuning. Using strictly resonant excitation, it was shown that one can reach $p_{\text{occ}} = 98$ % in similar devices [Maillette de Buy Wenniger et al., *Optica Quantum* 2, 404–412 (2024)]. To do so, we shall use polarized micropillar cavities and exploit a charged QD as in ref. [7] pushing the probability to emit a photon in a defined mode around $p_{\text{occ}} * \beta \approx 90$ %.

Improvements of the integration and operation system:

- The spectral laser filtering setup implemented in the present work shows a transmission of 73 %. We have recently developed a setup with polarisation filtering for resonant excitation with $\eta_{\text{setup}} = 90$ %.

- In the current work, the splicing between the high NA fibre and 780HP fibre results in a 10 % loss. We anticipate that advanced splicing technique should allow reaching $\eta_{\text{splice}} = 95$ %.

- Finally, the gap of $3 \mu\text{m}$ was chosen to ensure that any direct contact with the micropillar itself is avoided, even when exerting a force on the supporting stamp and after thermal contraction at cryogenic temperatures. However, this choice is too conservative; our recent study shows that we are now in position to have the fibre in direct contact with the micropillar. This would allow to go from 71 % coupling to $\eta_{\text{fibre-coupling}} = 96$ %. Note that putting the fibre in contact with the micropillar strongly reduces any effect of wavefront distortion.

Overall, we expect, using the same technology for the source and fibre-pigtailing considering already achieved setup efficiency for resonant fluorescence to reach $\eta_{\text{top}} * p_{\text{occ}} * \beta * \eta_{\text{setup}} * \eta_{\text{splice}} * \eta_{\text{fibre-coupling}} = 68.7$ % at the threshold for MBQC considering record value of single photon detectors above 95%.

Changes in the text:

Numerical simulations show that the outcoupling efficiency η_{top} of our micropillar is currently limited by a deviation from the ideal verticality by one or two degrees, leading to the current value of $\eta_{\text{top}} \approx 65$ % when $\eta_{\text{top}} = 93$ % is achievable.

*The use of near resonant excitation, while providing strong stability results occupation probability p_{occ} of the QD target state estimated to be 70 % for the detuning of -0.8 nm used here. Turning to resonant excitation scheme would allow $p_{\text{occ}} = 98$ % as measured in ref. [50] which could be combined with a polarised Purcell effect as in reference [7] to push the probability to emit a photon in a defined mode around $p_{\text{occ}} * \beta \approx 90$ %.*

Regarding the pigtailling process, it is possible to engineer the resin stamp on the fibre tip to approach it even closer and operate the system in contact with the micropillar to reach $\eta_{\text{fibre-coupling}} = 96$ %. An advanced fibre splicing should allow reaching $\eta_{\text{splice}} = 95$ % with optical setup for resonant excitation polarisation filtering now approaches efficiency around $\eta_{\text{setup}} = 90$ %. All these improvements combined should allow reaching the threshold of 2/3 for measurement-based quantum computation [51] considering the detector efficiencies currently above 98 % [52].

3. The authors mention that another excitation technique is needed to increase M significantly - do they have data on achievable through-fiber cross-polarization contrast ratios?

Answer: We think there has been a misunderstanding. In the manuscript we associated the use of another excitation technique to the improvement of the brightness as discussed above, not the indistinguishability (M).

We consistently observe the same indistinguishability with LA-phonon-assisted and resonant excitation. The difference between the two excitation techniques lies in the occupation probability that currently limits the source efficiency in LA excitation.

The feasibility of the implementation of laser suppression by cross-polarisation in a fibred setup, which is needed for achieving a high extinction ratio in resonant excitation, has recently been demonstrated both internally at Quandela and in another lab where they used a commercially available Quandela source, with a single-photon purity of $g^{(2)}(0) < 1\%$.

4. About the brightness value: "From the output single-photon rate and the laser repetition rate, we calculate a fibred brightness of $20.8 \pm 0.8\%$ ". How does this compare to the other mentioned numbers, it is yet not fully clear to me? Is the brightness measured in the 780HP fiber, is filtering taken into account? In the efficiency budget below, I still get 0.31 and not around 0.2:

0.7 excitation probability

0.75 cav-fiber coupl

0.66 filtering

0.9 splice

=0.31

Answer: We are sorry for not providing enough details to allow understanding of our total loss budget. This has now been clarified in the reviewed version of the article, including a detailed account of the brightness calculation in the methods and a clearer definition of where the brightness is measured.

The fibred brightness (now renamed brightness at the output fibre for clarity) is measured at the output fibre. This is not to be confused with the collection fibre before filtering. The detailed setup schematic is also now detailed in figure 4a.

Starting from the first-lens brightness at the output of the micropillar, the brightness at the output fibre includes the losses due to the coupling to the high-NA fibre, the splicing to the 780HP fibre, and the filtering stage for suppressing the excitation light.

The brightness in the output fibre is hence measured at the output of a 780HP fibre, after the final filtering stage. This number is thus the only relevant number for the end user.

Changes in the text: We provide all the details to understand the loss budget in the methods as well as in the figure legend of figure 4a where we indicate the transmission of various parts of the setup.

	Free space	Fibre-pigtailed
Detected countrate (MHz)	5.80 ± 0.05	4.78 ± 0.05
APD quantum efficiency	0.28 ± 0.01	0.29 ± 0.01
Laser repetition rate (MHz)	79.42 ± 0.01	79.21 ± 0.01
Brightness (output fibre)	$0.261 \pm 0.010^*$	$0.208 \pm 0.008^*$
Filtering setup transmission	0.68 ± 0.02	0.66 ± 0.02
High-NA to 780HP splice	–	0.90 ± 0.02
Micropillar-fibre coupling	0.82 ± 0.02	$0.75 \pm 0.05^*$
Brightness (first lens)	$0.468 \pm 0.025^*$	

*calculated value

5. I first was wondering why the cryocoolers have been exchanged twice (Fig. 3) - has the device really been completely remounted into different cryocooler cryostats?

Answer: The source has indeed been studied in three different compact cryocoolers, as part of the overall validation process and the transfer of the same source to different experimental facilities.

The installation procedure from one cryocooler to another only requires cutting the fibre and performing a new splice in the new cryocooler.

Changes in the text: This procedure is now explained in the main text to avoid possible confusions.

6. Why has the gap been chosen to be as big as 3.5um, leading to reduced coupling?

Answer: See response to questions 1,2 above.

Reviewer #3 (Remarks to the Author):

A quantum dot (QD) in a micropillar is a well developed single photon source (SPS). The present paper describes a technique to create a fiber-pigtailed SPS. The SPS has excellent g_2 and HOM metrics. The efficiency ("fibred brightness") is 20.8%. This efficiency is much higher than previous fiber-pigtailed SPSs but considerably lower than state-of-the-art SPSs. The fiber-pigtail is carried out by positioning a fiber using the cavity modes as a guide. The fiber is locked in place at room temperature and the positioning is maintained on cooling down. This is nice I think.

Answer: We thank the reviewer for this constructive report that allows us to improve our manuscript. Please find below our responses to all comments.

Important comments:

1. The paper focusses exclusively on engineering of micropillars. SPS is a broad field, but the engineering in the paper has a narrow scope. Are there wider consequences of the work?

Answer: We had not addressed this important aspect in the original version. We therefore thank the reviewer for this valuable comment that clearly highlights the impact of our work beyond our specific use case.

Indeed, our technique can be applied to other solid-state emitters in optical micro- and nano-structures with out-of-plane optical emission, such as for example bullseye cavities and nano-trumpets/wires. As long as a very good mode overlap is ensured with a single mode fibre, which has been verified with the above-mentioned cavities, the photon extraction should be very efficient, while at the same time the absence of any direct contact between the on-chip nanostructure and the fibre tip avoids damage on the sample side.

Changes in the text:

It is worth noting that the presented fibre-pigtail method avoids contact with the source while proving stable assembly over extended periods of time and cooling cycles.

This feature makes our approach particularly suited for fragile photon-extraction structures.

The present pigtail technique is thus compatible with different cavity designs with emission out of plane such as bullseyes, nanotrumpets, and nanowires among others, which have all allowed to obtain efficient single photon collection [53– 55].

2. The long-term stability Fig. 5 is good but Fig. 5 probes only noise at very low frequencies. The cryocooler produces mechanical noise (maybe also electrical noise) at specific frequencies. How does the SPS respond at these specific frequencies? For instance, is there any change in flux at the main ~ 1 Hz frequency of the cryocooler? The authors mention "to eliminate vibrations" - what's the evidence that this is achieved?

Answer: We are sorry if our sentence was unclear. When stating that "The mechanical stability obtained in this way is needed to prevent sliding of the fibre tip during cooldown and to eliminate vibrations", we meant "relative vibrations between the micropillar and the fibre" both during the cooldown and when operated at cryogenic temperature.

Changes in the text: We have modified the sentence accordingly which now reads:

“The mechanical stability obtained in this way is needed to prevent sliding of the fibre tip during cooldown and to eliminate relative vibrations between the micropillar and the fibre.”

Answer: This marginal relative movement is evidenced when monitoring the visibility of the fundamental and second mode of the cavity in reflectivity spectra, which allows to detect relative displacement of 200 nm as shown in figure 2 e,f. Moreover, when monitoring the QD emission intensity in a free space measurement, we can estimate that a reduction of the photon flux of 3% would correspond to a relative displacement between the fibre and the micropillar of 100 nm.

Based on these considerations, we attribute the small variation of the photon flux to electrical noise that slightly modifies the QD-cavity detuning, hence the brightness and indistinguishability.

Changes in the text:

We estimate that a variation of single photon rate by less than 3 % would correspond to a relative displacement of the fibre-micropillar alignment by 100 nm. We thus attribute the observed deviations from the maximum value mainly to instabilities in the electrical environment of the QD which slightly modifies the QD-cavity detuning hence the indistinguishability.

Answer: To rule out any effect linked to the cryocooler, we show in the figure a more resolved timetrace (each datapoint of the flux shown in fig 5a was averaged for 2 seconds with 10 points per second). From this data, we do not observe variations of the countrate at the frequency of the cryocooler.

3. HOM - results are impressive but are reported only on interfering successive photons, i.e., photons created 12 ns apart in time. For any multi-photon applications involving demultiplexing the HOM needs to be maintained for photons created far apart in time. What can the authors say about this? And HOM is still less than Ref. 3 [Somaschi et al.] - why?

Answer: We did not perform any long-delay HOM interference measurements on the source presented in the present paper. On another fibre-pigtailed source with $M = 92.9\%$ we observed a reduction to 85.1% after $0.75\ \mu\text{s}$ (which corresponds to photons separated by 60 other photons), similar to standard observations in micropillar sources [Optica 3, 433-440 (2016)].

Regarding the comparison with Ref. 3, indistinguishability values as high as $M = 99.56\%$ were obtained with an etalon for additional spectral filtering of the phonon sidebands. As shown in Grange et al., ref. 49, few percent difference in indistinguishability is expected and observed when comparing the indistinguishability of QD-micropillar sources with similar quality factors with and without spectral filtering of the phonon sideband.

Changes in the text:

we obtain an indistinguishability $M = 97.5 \pm 0.1\%$. Such value is on par with the highest values reported for QD-micropillar cavities [Somaschi 2016] considering that the rather broad (0.8 nm) spectral filtering hardly suppresses the phonon sidebands contributions. This is confirmed by the observation of an increased indistinguishability from $M = 94.3$ in the free-space characterisation setup, which we attribute to the lower working temperature of 2.4 K for the cryocooler hosting the fibre-coupled device compared to 5 K for the low-vibration cryocooler used for the free-space characterisation. While strongly suppressed by the acceleration of the zero-phonon line (see ref. [48, 49]), the residual contribution of the phonon sidebands is further reduced at lower temperatures.

Minor comments:

1. Why use "brightness" for 20.8%? I would suggest that 20.8% is better described as an overall efficiency or something like that. What does "brightness" mean in this context? I would say that it's efficiency x repetition rate.

Answer: We use the term "brightness" in continuity with the definition of first-lens brightness. Both are defined as the probability (thus unitless) of having a photon at some defined point of the system for each excitation event. This is corrected for the $g^{(2)}(0)$ of the source to exclude the extra photons and keep the metric useful for a single-photon source. It is equivalent to the overall efficiency, but we think this wording helps in understanding where the efficiency or brightness is evaluated and makes the comparison between different works easier.

Changes in the text: The definition of brightness has been improved in the text and a figure 4a has been added to show where each quantity is measured in our experimental setup.

2. does fiber act as a waveplate - does a linear polarization at input give a linear polarization at output?

Answer: The fibres used in our experiments are not polarisation-maintaining. Therefore, a linear polarisation at the input is both rotated and made elliptical by the transmission through the fibre. To compensate for this effect, we use a quarter- and a half-waveplate in the filtering stage of the setup to compensate for this unwanted polarisation rotation (see figure 4a). For the output fibre, three paddles are used to control the polarisation.

Changes in the text: We have added this detail in figure 4 caption.

3. "fibred brightness" implies that there is a verb "to fiber" yet "fiber" is a noun.

Answer: Our initial choice of defining "fibre brightness" was motivated to avoid the possible confusion with the "in-fibre brightness" defined in several previous articles as the brightness in the fibre, just after collection from the emitter (and not after filtering of the excitation light).

Changes in the text: We now describe the two brightness values in a clearer way and avoid the use of “fibred”. Specifically, we introduce

- *the first lens brightness B_{fl} as the probability of collecting a photon per excitation pulse at the output of the micropillar in a 0.7 NA lens*
- *The brightness at the output fibre $B_{out-fib}$ as the probability of collecting a photon per excitation pulse at the output of the system after collection and filtering*

4. $F_p=13$ implies a beta-factor $F_p/(F_p+1)=93\%$, setting (I think) the maximum first-lens efficiency. What are the loss mechanisms leading to 46.8%?

Answer: Considering that there would be no suppression of spontaneous emission by the micropillar structure, the reviewer estimation is correct. We describe all the other limitation of our system in the response to reviewer 2 - questions 1&2.

5. why splice the high-NA fiber to a regular fiber? This introduces a mode-mismatch and loss.

Answer: We chose a high-NA fibre that optimizes the mode matching with the micropillars with diameters around $3\ \mu\text{m}$. To ensure decent mode matching directly with a standard 780HP fibre, one would have to increase the size of the micropillar by several micrometres, thus significantly increasing the number of mirror pairs in the pillar to maintain a high enough Purcell factor.